# Crosstalk between Mitochondria and Cytoskeleton in Cardiac Cells

**DOI:** 10.3390/cells9010222

**Published:** 2020-01-16

**Authors:** Andrey V. Kuznetsov, Sabzali Javadov, Michael Grimm, Raimund Margreiter, Michael J. Ausserlechner, Judith Hagenbuchner

**Affiliations:** 1Cardiac Surgery Research Laboratory, Department of Cardiac Surgery, Innsbruck Medical University, 6020 Innsbruck, Austria; michael.grimm@tirol-kliniken.at; 2Department of Paediatrics I, Medical University of Innsbruck, 6020 Innsbruck, Austria; michael.j.ausserlechner@i-med.ac.at; 3Department of Physiology, School of Medicine, University of Puerto Rico, San Juan, PR 00936-5067, USA; sabzali.javadov@upr.edu; 4Department of Visceral, Transplant and Thoracic Surgery, Medical University of Innsbruck, 6020 Innsbruck, Austria; raimund.margreiter@tirol-kliniken.at; 5Department of Paediatrics II, Medical University of Innsbruck, 6020 Innsbruck, Austria

**Keywords:** heart, cytoskeletal proteins, mitochondria, energy metabolism, mitochondrial interactions, plectin, tubulin beta, signaling

## Abstract

Elucidation of the mitochondrial regulatory mechanisms for the understanding of muscle bioenergetics and the role of mitochondria is a fundamental problem in cellular physiology and pathophysiology. The cytoskeleton (microtubules, intermediate filaments, microfilaments) plays a central role in the maintenance of mitochondrial shape, location, and motility. In addition, numerous interactions between cytoskeletal proteins and mitochondria can actively participate in the regulation of mitochondrial respiration and oxidative phosphorylation. In cardiac and skeletal muscles, mitochondrial positions are tightly fixed, providing their regular arrangement and numerous interactions with other cellular structures such as sarcoplasmic reticulum and cytoskeleton. This can involve association of cytoskeletal proteins with voltage-dependent anion channel (VDAC), thereby, governing the permeability of the outer mitochondrial membrane (OMM) to metabolites, and regulating cell energy metabolism. Cardiomyocytes and myocardial fibers demonstrate regular arrangement of tubulin beta-II isoform entirely co-localized with mitochondria, in contrast to other isoforms of tubulin. This observation suggests the participation of tubulin beta-II in the regulation of OMM permeability through interaction with VDAC. The OMM permeability is also regulated by the specific isoform of cytolinker protein plectin. This review summarizes and discusses previous studies on the role of cytoskeletal proteins in the regulation of energy metabolism and mitochondrial function, adenosine triphosphate (ATP) production, and energy transfer.

## 1. Introduction

Cells are highly organized units with multifaceted functional and structural interactions between various subcellular systems. A large number of studies provides strong evidence that elucidating individual organelles alone is not sufficient, and only systemic approaches must be applied for understanding intracellular signaling pathways and crosstalk between subcellular organelles. This may involve a “systems biology” approach and combinations of several most modern technologies such as genetic manipulations, live cell imaging, mathematical modelling, etc. In high oxygen consuming organs like the heart, energy supply (ATP) is provided by mitochondria in the reactions of oxidative phosphorylation (OXPHOS). Notably, mitochondria actively interact with other subcellular organelles and systems like cytoskeleton and sarcoplasmic reticulum (SR) [1,2,3,4,5,6,7,8,9,10,11,12]. Many cytoskeletal elements play a vital role in the structural and functional organization of mitochondria, including mitochondrial shape and morphology, dynamics, motility, and mitosis [13,14,15,16,17]. Most importantly, the interaction of mitochondria with some cytoskeletal proteins and their connections to voltage dependent anion channel (VDAC) can be involved in the coordination of mitochondrial function [18,19,20,21,22,23] (Figure 1). In the heart, mitochondrial bioenergetics and oxygen consumption are linearly dependent on the cardiac contractile activity [24,25] at rather stable concentration of the main mitochondrial regulator adenosine diphosphate (ADP), which is a central element in mitochondrial physiology. The exact mechanisms of how mitochondria precisely respond to the heart energy demand remained unknown for a long time and require further investigations. A growing body of evidence shows that the cells contain intracellular metabolic micro-compartments provided by multidirectional mitochondrial interactions with other subcellular organelles and macromolecules, in particular, specific cytoskeletal proteins [26,27,28,29,30,31,32,33,34]. In this review, we summarize and discuss previous studies that provide strong evidence for the role of cytoskeletal proteins, in particular, tubulin beta-II and plectin 1b, in the regulation of mitochondrial bioenergetics and energy fluxes via the energy-transferring supercomplex VDAC-mitochondrial creatine kinase (MitCK)-ATP-ADP translocase (ANT) under physiological and pathological conditions.

## 2. Historical Retrospective

The heart is a high oxygen consuming and ATP demanding organ with a large number of mitochondria that occupy ~30% of cardiac cell volume. Besides supplying the cardiac tissue with ATP, mitochondria play an important role in cell signaling, differentiation and growth, as well as in the maintenance of the cellular redox system, ion homeostasis, and cell death, actively communicating with other cellular systems like SR and cytoskeleton. The presence of micro-compartmentation of ATP and ADP (i.e., their high local concentrations at mitochondria and close to myofibrils) was evident from the observations that cellular bulk concentrations of ATP and ADP are relatively constant, independently of changes in heart workload. Interestingly, the total ischemia or anoxia quickly stops heart contractility while cellular bulk ATP concentration decreases by only ~5% under these conditions. Furthermore, the free cellular concentration of ADP in the heart (usually ~20 µM) cannot be higher than 50 µM, otherwise it will eventually lead to the increased left ventricular end diastolic pressure and thus, to the cardiac rigor super-contracture. On the other hand, the full activation of mitochondrial respiration requires at least 250–300 µM of ADP in isolated mitochondrial preparations. The detailed mechanisms of precise matches and synchronizations of mitochondrial respiratory function and heart contractility (excellently tuned cellular energy production and demand) still remain unclear and are under active investigation by several groups [27,28,30,31,32,33,34]. Apparently, mitochondria–cytoskeleton interactions play a certain role in these crosstalk mechanisms.

The pioneering work of Denton and McCormack in the 1980s [35] followed by other studies [36] proposed that intramitochondrial Ca^2+^ can activate the dehydrogenases involved in the tricarboxylic acid cycle and lead to upregulation of electron transfer chain (ETC) and OXPHOS, associated with high ATP production [35,36]. This metabolic regulation of mitochondrial bioenergetics by Ca^2+^ is known as a “parallel activation model” in the heart. According to this theory, increased cardiac contractile function and energy demands both are achieved by the increased cytosolic and mitochondrial Ca^2+^ with the involvement of several Ca^2+^ carriers. As a result, increased matrix Ca^2+^ stimulates mitochondrial dehydrogenases, mitochondrial function, and ATP production to match the increased energy consumption by myofibrils. Notably, these processes are shown to be strongly tissue specific [36].

For a long period of time, mitochondrial function was investigated mostly using isolated mitochondria in vitro. The apparent Michaelis constant (appKm) for the main mitochondrial substrate ADP in Michaelis–Menten equation is an important parameter of mitochondrial respiratory function, which can be obtained from the respiratory ADP kinetics. This parameter reflects the affinity of mitochondrial respiration to ADP and the permeability of the outer mitochondrial membrane (OMM). For many types of isolated mitochondria, this parameter was in a range of 10–30 µM [37,38]. These types of studies, however, resulted in the loss of the mitochondria-cytoskeleton interactions that are important for the control of metabolites transport in mitochondria, and for the regulation of the mitochondrial respiratory function. In vivo or in situ measurements of mitochondrial respiration (e.g., in permeabilized cells) could also be essential [39,40].

Kummel [41] and several other researchers [42,43,44,45] discovered the functional differences between isolated mitochondria in vitro and non-isolated mitochondria in situ (in permeabilized cardiac cells or muscle fibers). It has been found that appKm for external ADP, which is important for regulation of mitochondrial respiratory function, is significantly different in vitro and in situ mitochondria [42,44,45]. Therefore, instead of isolated mitochondria, myocardial fibers or cardiac cells permeabilized by digitonin or saponin were effectively used for the characterization of mitochondrial energetics in studies from the Saks group among others [42,43,44,45]. This approach allows avoiding mitochondrial isolation, and therefore has a number of serious advantages, most importantly, preserving mitochondrial contacts with other subcellular structures and systems, including the cytoskeleton [39]. Surprisingly, the appKm for ADP for mitochondria in situ was found to be about 300–400 µM [43,44,45,46], which is very different compared to isolated mitochondria. Importantly, the mild proteolytic treatment, e.g., with trypsin, significantly decreased appKm for ADP in permeabilized preparations almost to the value of in vitro, isolated mitochondria [26].

All these observations pointed to the involvement of cytoskeletal proteins as primary candidates in the control of mitochondrial respiratory function. Imaging analysis (fluorescence and immunofluorescence confocal microscopy) of cardiac cells and muscle fibers by using specific mitochondrial markers and various antibodies revealed full colocalization of mitochondria with cytoskeletal protein tubulin beta-II, suggesting its structural and functional interactions with mitochondrial VDAC [22,32,46,47]. Notably, in HL-1 cardiac cells that are devoid of tubulin beta-II, mitochondrial respiratory behavior and sensitivity to ADP (appKm) were similar to that of isolated mitochondria [47]. More recently, respirometrical and imaging analyses demonstrated that plectin 1b isoform is associated with mitochondria [48], which like tubulin beta-II, can also control the permeability of the OMM and thereby, modulate mitochondrial function. In favor of this, cardiac and muscle tissues from plectin 1b knockout mice showed severe mitochondrial changes and reversed sensitivity to ADP as evidenced by decreased appKm [48].

## 3. The Role of Cytoskeleton in the Mitochondrial Intracellular Organization, Shape Morphology and Dynamics

In various cells, mitochondria are associated with the three major cytoskeletal structures microtubules, intermediate filaments (IFs) and microfilaments [49,50,51,52,53,54,55]. It is known that specific cytoskeletal proteins are central for the mitochondrial morphology, dynamics, motility, intracellular traffic and mitosis [2,6]. Mitochondria can be associated with the actin-network and could be either anchored on cytoskeletal filaments or shaped by the forces (mechanical factors) generated by actin (see references in Section 8 “Cytoskeletal-Mitochondria Interactions in Pathology”). Microtubules are considered to be primary tools for mitochondrial transport [55,56]. However, actin is also required for short-distance mitochondrial activities as well as for the immobilization (anchorage) of mitochondria that may be important for holding these organelles at sites of higher energy demands. Moreover, some mutations in actin or actin-binding proteins may affect mitochondrial mechanisms leading to cell death [53]. Actually, mitochondria-actin interactions have been shown to be involved in apoptosis.

In cardiomyocytes, accumulation of intermyofibrillar mitochondria is observed at the vicinity of t-tubular network and separated by sarcomeric Z-lines in sarcomeres, that can be labeled by α-actinin immunofluorescent staining (Figure 2A–C) [32,33,48]. Many specific proteins that regulate mitochondrial intracellular localization, organization, shape/morphology, dynamics, and motility have been discovered [13,14,15,16,17,57]. Mitochondrial shape under physiological conditions usually needs the attachment of the organelles to cytoskeleton elements as the internal scaffolding system. Various mitochondria-shaping proteins have been identified and significant alterations in mitochondrial morphology and/or intracellular organization were observed in specific mutants [53]. Several special proteins can be responsible for the control of the mitochondrial shape through interactions of mitochondria to the cytoskeleton, while others can be responsible for the formation of connections between the OMM and inner mitochondrial membrane (IMM). The formation of the regular tubular shape of mitochondria normally needs several OMM proteins such as Mmm1p, Mmm2p, Mdm10p and Mdm12p as well as the IMM proteins Mdm31p and Mdm32p [57,58,59,60,61,62]. In mutants lacking any of these proteins, mitochondrial tabulation and elongated and branched shapes of tubules may disappear, and mitochondria can be then organized into big clusters of spherical shape. It has been shown that Mmm1p, Mdm10p, and Mdm12p can form the specific MMM complex, which, in cooperation with Mmm2p, Mdm31p, and Mdm32p proteins, stimulates formation of tubular mitochondria. This complex can be involved in the attachment of mitochondria to actin, interacting also with other cytoskeletal scaffolding systems. Mitochondrial morphology, the IMM and cristae shapes of mitochondria can also be regulated by Mdm33p, Gem1p, mitofilin [63,64], and ATP synthase [65].

In many cell types (mammals, yeast, etc.), the two opposing mitochondrial fission and fusion processes are regulated by various specific proteins [17,66,67,68,69,70,71,72,73,74,75]. Mitochondrial fusion and fission are regulated by the dynamin family GTPases [17]. Dynamin-related protein 1 (Drp1 or DLP1) [70,71] and mitochondrial fission 1 protein (Fis1/hFis1) participate in mitochondrial fission [72], whereas mitofusin 1 (Mfn1) and 2 (Mfn2), and optic atrophy protein 1 (OPA1) in mammalian cells regulate mitochondrial fusion [66,67,68,69,74]. Importantly, both cytoskeletal microfilaments and microtubules can be involved in the recruitment of Drp1 to mitochondria [70]. Fission–fusion shifts can frequently occur under various stressful conditions (oxidative stress, ischemia-reperfusion injury, etc.) [69,75,76,77,78,79], representing also an early event in the mitochondria-dependent programmed cell death (apoptosis) [80,81,82]. Cardiac ischemia-reperfusion injury (IRI) and post-infarction heart failure has been shown to increase mitochondrial fragmentation due to alterations in the expression and post-translational modifications of mitochondrial fission-fusion proteins [77,79]. The interactions of mitochondria with the cytoskeleton can be critical for the accumulation of mitochondria in specific cellular regions and mitochondrial movement can provide a local energy production at sites of higher ATP demands [83,84,85,86]. Mitochondrial movement (transport along microtubules), well known in neurons, is based on the several specific motor proteins such as the kinesin family of mitochondria-bound proteins and on the interactions with other cytoskeletal microtubules-dependent proteins [84,85,86,87]. In contrast, in cardiac cells, mitochondria are strongly fixed between myofibrils (intermyofibrillar mitochondrial subpopulation), which is absolutely obligatory for the normal organ contractile function.

Myosin V additionally contributes to organelle transport along actin networks. It has been shown that several protein kinases as well as the phosphorylation of certain proteins of microtubules can be involved in mitochondria-cytoskeleton crosstalk through interactions with mitochondrial membranes. In addition, phosphatidylinositol 3-kinase signaling pathways are also important for motility of mitochondria [85,88]. The proteins that arrange a link such as motor molecules-mitochondria, motor-independent motilities, and anchorage of mitochondria at cortical sites provide a connection between mitochondria-cytoskeleton interactions and mitochondrial flexibility [88]. The proper coordination of the mitochondrial dynamics is important for normal functioning of mitochondria, and mutations in the proteins that control the mitochondrial dynamics result in human diseases [69,79]. Notably, mitochondrial morphology, intracellular arrangement, and specific proteins involved in the mitochondrial dynamics are extremely cell-tissue specific [89]. Finally, mitochondrial interactions with the cytoskeleton network are shown to be important not only for the control of their morphology, dynamics, and organization, but also for the regulation of the entire energy metabolism [90] and OMM permeability to metabolites [2], as well as overall mitochondrial physiology [91]. The entire cytoskeleton and specific cytoskeletal proteins can contact mitochondria to control the OMM permeability to ADP and regulate OXPHOS, the main function of mitochondria.

## 4. Cytoskeleton and Mitochondria-SR Interactions

Mitochondria play a central role in cell life and cell death and mediate a myriad of intracellular pro-survival and pro-death signaling pathways [92]. Several physiological mechanisms need precise interactions between various subcellular organelles, like plasma membrane, nucleus, mitochondria and SR. In myocardium, Ca^2+^ released from SR and Ca^2+^ cycling plays a fundamental role in the excitation–contraction coupling, as well as in the interactions of different cytoskeletal elements with mitochondria. Overall, the function of the heart, Ca^2+^ homeostasis, and excitation–contraction coupling vitally depend on the ATP production by mitochondrial OXPHOS. On the other hand, mitochondrial Ca^2+^ overloading can damage mitochondria, reducing ATP production, leading to ROS generation, oxidative stress and various cardiac injuries (for more details, see Section 8).

Some dense structures were frequently observed between the OMM and SR or T-tubules that can link these systems [3,93]. The communications and interactions of subcellular organelles are based on the vesicular trafficking and membrane contact sites important in Ca^2+^ homeostasis and lipid metabolism. These tight interactions and contacts permit cells and their specific compartments to adapt them to the different conditions [3]. It has been proposed that such contacts are essential for the transport of lipids (phospholipids) as well as for the overall cellular Ca^2+^ homeostasis via the complex formed by VDAC and inositol 1,4,5-trisphophate receptors, managing vital cellular processes like contraction, secretion, cell growth, proliferation, apoptosis, etc. [94]. Specific elements of cytoskeleton can be associated with L-type Ca^2+^ channel, and regulate its activity and mitochondrial function, mediating mitochondrial membrane potential [95].

It has been demonstrated that the distances between membranes of organelles assessed by electron microscopy are relatively small, and allow to create structural contacts between proteins of these membranes. This regulates organelle–organelle interactions, restructuring the mitochondrial morphology and network together with Ca^2+^ handling under physiological conditions. The membrane contact sites (mitochondria-associated membranes) occur in response to various mitochondrial or SR stresses (autophagy, apoptosis, inflammation) [96,97,98], as well as in several diseases associated with changes in mitochondrial dynamics machinery [99]. They may also play an important role in various diseases such as neurodegenerative diseases, diabetes, infection diseases and cancer [100]. Mitochondria-SR contacts have been shown to be involved in the mitochondria–cytoskeleton interactions, regulating mitochondrial dynamics, including mitochondrial fusion and fission processes [96]. On the electron micrographs, such contacts look like SR tubules closely faced to mitochondria.

## 5. Possible Role of the Intermediate Filaments Proteins Desmin and Vimentin in the Regulation of Mitochondrial Bioenergetics

Cardiac and skeletal muscle cells contain intracellular network that tightly regulates myofibrillar activity and maintains muscle contraction/relaxation. The synchronization of the basic contractile element, sarcomere, involves well-organized filament structures that include myosin (thick structure), actin (thin structure), nebulin, and titin [101]. It is connected to other subcellular organelles such as the nucleus and mitochondria. As a result, the multiorganelle network operates as a platform for general cellular integrity/stability, also governing mitochondrial function, shape, and intracellular organization. The contractile machinery represents a complex network, all three members of which (microtubules, IFs and microfilaments) are associated with mitochondria [49,50,51,52,53,54]. IFs are considered as the main protectors against various stresses such as oxidative stress, toxic injury, apoptotic stimuli, etc. [91]. They also play an important role in cell growth/differentiation, bioenergetics, cellular signaling and cells relocation. IFs have the ability to be polymerized and their mechanical properties and richness can change in response to pathological stimuli. IFs maintain the cell integrity and thus play an important role in protein targeting and inter-organellar interaction. Mitochondrial function and subcellular organization may be regulated by IFs proteins as shown for IFs desmin, vimentin and some other proteins [30,34,102,103]. Also, intracellular locations of Golgi can be regulated by IFs.

In the cell, the desmin cytoskeleton is responsible for the proper mitochondrial positioning and shape. It may also regulate the formation and stabilization of mitochondrial contact sites. Desmin is present in cardiac, skeletal and smooth muscle cells, in particular, in dense bodies, nearby the nuclei, close to the Z-line and costameres. It can be upregulated during muscle adaptations as well as in myopathies, muscle degeneration, and drug treatments [102]. Desmin has been suggested to participate in the regulation of myofibrillogenesis, mechanical support of the muscle cells, mitochondrial localization, gene expression and intracellular signaling. It can interact with actin, tubulin, plectin (cytolinker protein) and dynein (motor protein). IFs, like microtubules (see above), were suggested to have a significant impact on mitochondrial morphology, as well as cellular organization and functions in different mammalian cell types. Changes in their interactions can lead to various human diseases [104,105]. Several studies with desmin-deficient (desmin-null) mice [30,34,102] have demonstrated the importance of desmin in subcellular distribution and respiratory function of mitochondria. Ultrastructural studies of cardiomyocytes from desmin-null models showed mitochondrial proliferation that was elevated in response to increased workload. Cardiac and skeletal muscles of desmin-null mice exhibited significant changes in the morphology and intracellular organization of mitochondria [30,34].

Mitochondrial alterations in desmin-null muscles were associated with decreased maximal rate of respiration (ADP-stimulated rate of oxygen consumption). Also, the lack of the coupling between MitCK and ANT observed in desmin-null models [22,31,33,106,107,108] indicates alteration of intracellular energy transfer [22,31,33,92]. In addition, the decrease in mitochondrial respiration was associated with the decline of the appKm for ADP in permeabilized cardiac fibers of desmin-null mice. These data show that desmin can participate in the regulation of the mitochondrial VDAC, directly or via a desmin-associated cytolinker protein plectin. In contrast, mitochondrial function and appKm for ADP in permeabilized fibers from skeletal glycolytic muscles were not seriously affected in the absence of desmin [30]. Proteomic analysis of cardiac mitochondria isolated from desmin knockout mice has demonstrated alterations in various metabolic processes such as apoptic pathways and Ca^2+^ cycling. The changes in VDAC expression suggested a connection between the desmin-determined cellular organization and mitochondrial energy metabolism [30,34]. Cardiac and skeletal muscles of aggregation-prone desmin mutant L345P mice exhibited significant changes in morphology of mitochondria and Ca^2+^ handling. Al these studies proposed that desmin directly or indirectly can participate in the regulation of mitochondrial function.

Several studies suggested that the cytoskeletal IF protein vimentin, like dismin, can also regulate mitochondrial bioenergetics [103]. Like desmin, vimentin can interact with mitochondria [103,109,110] and modulate their shape/morphology, intracellular organization and dynamics [103]. Vimentin-null cells displayed lower mitochondrial membrane potential, which was recovered by adding of external vimentin [110]. The cytolinker protein, plectin, which is expressed ubiquitously, participates in mitochondria–vimentin interaction [109]. Its specific mitochondrial isoform plectin-1b has been suggested to bind vimentin to mitochondria. Thus, desmin, vimentin, and plectin-1b are critical for functional and structural interactions between the cytoskeleton and mitochondria that regulate mitochondrial function. It should be noted that the direct involvement of desmin and vimentin in mitochondrial bioenergetics and physiology is still debated.

## 6. The Role of Tubulin in the Regulation of Mitochondrial Bioenergetics and Metabolism

In oxidative muscles, mitochondria are organized into functional complexes with myofibrils and SR, and create specific intracellular energetic units [32,33,111,112]. Energy crosstalk within these units provides facilitated diffusion of ADP, metabolic micro-compartmentation and channelling by the local energy transfer network [31,33] which includes creatine kinase and adenylate kinase. The microtubules are mostly composed of tubulin; their assembly and function are regulated by the microtubule associated proteins kinesin and dynein. In many cell types and tissues, mitochondria typically show an intracellular distribution matching the microtubular organization [32,113]. 

In the heart, tubulins form a network which, together with plectin, desmin, and microfilament proteins (actin), creates a precise structural organization of cardiac cells. This organization is essential to sustain the cardiac contractile function, as well as for the regulation of energy supply and demand [19,21,22,23,101,113,114,115]. It is known that tubulin in vivo is dynamic, undergoing assembly/disassembly processes due to interchanges between its subunits. The microtubule units are formed by alpha and beta tubulin heterodimers [116]. In cardiomyocytes, about 70% of total tubulin is present in the polymerized form as microtubules, whereas 30% occurs as a non-polymerized cytosolic heterodimeric protein [117,118,119]. Many chemical agents that depolymerize microtubules can significantly change mitochondrial intracellular organization [53,113]. Similar alterations in mitochondrial localization and motility were also found in cases of actin-encoding gene mutations demonstrating a possible role of the actin cytoskeleton [55,56]. Interestingly, after the complete dissociation of the microtubular system by colchicine, tubulin is still present in permeabilized cardiomyocytes, possibly due to the association with other cytoskeletal elements [2,120]. In 1990, Saetersdal et al. [114], for the first time, reported a possible connection of β-tubulin isotype to the OMM using microscopy and immunogold labelling of cardiac muscle cells. Furthermore, immunoprecipitation analysis has demonstrated direct association of tubulin with mitochondrial VDAC [121], confirming earlier suggestion of direct interconnections between microtubules and OMM [21]. Moreover, it was found that the addition of isolated dimeric tubulin induces closed state of VDAC, restoring the low permeability of the OMM, thereby, increasing appKm for ADP [19,20,121,122,123,124].

The ANT is less accessible to externally added ADP in permeabilized cardiac cells or oxidative muscle fibers than that in isolated mitochondria [40,41,42,43] (see Section 2). Regulation of the OMM permeability by VDAC channelling has two major functions. First, it controls mitochondrial respiration and energy transfer from energy source (mitochondria) to different sites of energy utilization in the cytoplasm. Numerous metabolites, such as respiratory substrates, ADP, and Pi, enter mitochondria only through VDAC. On the other side, high-energy phosphates, mostly ATP and phosphocreatine are channelled out through the VDAC to drive cellular energy transfer. Control of energy fluxes through VDAC is regulated by tubulin beta-II bound to VDAC (Figure 3). Second, VDAC becomes a channel for release of pro-apoptotic factors from mitochondria to the cytoplasm in response to apoptotic stimuli. Both tubulin beta-II and plectin can control VDAC permeability and therefore energy and metabolic fluxes of ATP, ADP, creatine (Cr) and phosphocreatine (PCr) (see Figure 3).

The significant role of tubulin in the modulation of mitochondrial VDAC has been extensively analysed during the last two decades [18,19,20,23,123,124,125]. It was found that the addition of αβ-tubulin to isolated cardiac mitochondria in vitro, at concentrations below the value critical for the tubulin polymerization, significantly increased appKm for ADP in ADP kinetics of mitochondrial respiration to the value of in situ mitochondria, discovered in the permeabilized preparations, thus demonstrating restricted (decreased) ADP availability to ANT [19,124]. It was also shown that the addition of αβ-heterodimeric tubulin to reconstituted, purified VDAC (three isoforms of VDAC 1, 2, and 3) provoked reversible transition to its closed state, limiting mitochondrial ADP or ATP fluxes [18,19,20,122]. These findings suggest a mechanism of regulation of mitochondrial energetics, governed by VDAC and tubulin at the mitochondria-cytosol interface. Depending on the applied voltage and phosphorylation state of VDAC, the low concentrations of dimeric tubulin may activate reversible obstruction of VDAC incorporated in the artificial phospholipid membrane. Analysis of the tubulin-closed state demonstrated that it can carry small ions, but is impermeable to ATP, ADP and other mitochondrial metabolites. All these observations were then confirmed in isolated mitochondria or human hepatoma cells. Tubulin–VDAC interactions require a specific structure of the anionic C-terminal tail of tubulin.

Altogether, the results of previous studies suggest cytoskeletal protein tubulin as an important player in the regulation of VDAC states and OMM permeability (permeability restrictions created by interactions of VDAC in OMM with tubulin) in the mechanisms of mitochondrial energetics regulations [19,31,47]. The functional VDAC–tubulin interactions can be either direct or indirect, via cytoskeleton proteins such as specific isoforms of plectin. The interaction between VDAC and tubulin can be affected by isoform patterns of both tubulin and VDAC, as well as by their post-translational modification (e.g., phosphorylation). The main differences between distinct isotypes of tubulin are located in the C-terminal residues (called also isotype defining region). The C-terminus can be a target for various microtubule associated proteins (MAPs) [125,126,127] and the differences between multiple interactions of tubulin with various cellular systems can be determined by the composition of the C-terminus. In addition, MitCK, which has been shown to tightly interact with VDAC, can act as an important regulating factor in these interactions. Notably, the ANT-MitCK pathway (phosphocreatine shuttle) can be active in oxidative but not in glycolytic muscles and cancer cells [47,128]. It was suggested that tubulin beta-VDAC interactions participate in the modulation of cellular energy metabolism in cancer switching it from the oxidative phosphorylation mode to more glycolytic phenotype known as the Warburg effect [129]. This phenomenon has recently received renewed interest [130,131,132].

Fluorescent confocal imaging can not only visualize mitochondrial intracellular arrangement, morphology, dynamics, networks, and heterogeneity, but also quantitatively analyze mitochondrial functional parameters, like redox state, membrane potentials and Ca^2+^ levels [133,134,135]. Moreover, the combination of live mitochondrial imaging in cells or muscle fibers, together with immunofluorescence visualization and immunoblotting of cytoskeletal proteins allows to analyze structural relationships between these structures. The imaging analysis of the intracellular distribution of tubulin isoforms in cardiac cells by immunofluorescence confocal microscopy has discovered a regular arrangement of tubulin beta-II (Figure 4). Most importantly, double imaging analyses by fluorescence and immunofluorescence microscopy demonstrated clear co-localization of tubulin beta-II with cardiac intermyofibrillar mitochondria [8]. Tubulin beta-IV demonstrated an organization in branched network while tubulin beta-III was localized close to Z-lines, and tubulin beta-I was diffusely distributed [8]. The colocalization of tubulin beta-II with mitochondria suggested its functional and structural interaction with mitochondrial VDAC [8,22,32]. Interestingly, permeabilized HL-1 cells with cardiac phenotype demonstrated mitochondrial parameters and appKm for ADP very similar to that of isolated mitochondria that indicates a high open state of VDAC. This parameter was very different from that found in adult cardiomyocytes or H9c2 cardioblastic cells [136,137]. The absence of tubulin beta-II in HL-1 cells shows the importance of tubulin beta-II in the control of the OMM permeability and mitochondrial function through regulations of VDAC open-close states [22,32]. The absence of tubulin beta-II and the presence of only β-IV-tubulin, β tubulin I and III can be explained by cancerous phenotype of these cells. HL-1 cells originate from mouse atrial cardiomyocytes and, apparently, are more dependent on glycolytic rather than mitochondrial energy production. Another important characteristic of HL-1 cells is the lack of MitCK, and therefore phosphocreatine-mediated energy transfer [47]. Accordingly, functional analysis of permeabilized cardiomyocytes and HL-1 cells such as ADP-kinetics, and stimulatory effects of creatine and glucose on mitochondrial respiration rates revealed dramatic differences. In HL-1 cells, the appKm for ADP was the same (~20 µM) as for isolated in vitro mitochondria [136]. All these findings show associative link between the structural (presence or absence of tubulin beta-II and MitCK) and functional (e.g., appKm for ADP) features in different primary cells and cell lines.

The association of tubulin beta-II with the OMM, when co-expressed with MitCK, may specifically limit the permeability of VDAC for adenine nucleotides, resulting in the formation of adenine nucleotides (ADP, ATP) micro-compartmentation in the mitochondrial intermembrane space. Thus, tubulin beta-II can participate in the control of VDAC, permeability of the OMM, and in the control of metabolic energy and metabolic fluxes (ATP, ADP, PCr, Cr, Pi) via the VDAC-MitCK-ANT supercomplex (Figure 3), thereby controlling cellular energy production and energy transfer in cardiac and oxidative muscle cells (Figure 3 and Figure 5A). This supercomplex, localized at contact sites of two mitochondrial membranes, is a key structure of a specific pathway for the effective energy transport from mitochondria to cytoplasm, as well as for the local regeneration of ATP at sites of energy utilization including myofibrils, SR, anabolic processes, and active transport (via various pumps) across the sarcolemma membrane [32,33,111,112].

## 7. Possible Role of Plectin in the Control of Mitochondrial Intracellular Organization and Function

Previous studies demonstrated that the proper cell architecture and subcellular organization in cardiac cells are strongly dependent on plectin and desmin, and proposed that plectin can use desmin in cellular signaling [138]. In muscles, four isoforms of plectin were found that possess different cellular anchoring functions: they control the structure, organization, and stability of the cells [138,139]. Plectin 1f connects desmin to the sarcolemma whereas plectin 1 binds desmin to the nucleus. Plectin 1d connects desmin IFs to the Z-disk in cultured myotubes and plectin 1b, a mitochondrial isoform of plectin can directly interact with VDAC [115,140,141]. Plectin 1b was shown to be inserted into the OMM with the exon 1b encoded N-terminal sequence, which operates as an anchoring and mitochondria-targeting indicator. The disruption of Z-disks and costameres connections leads to the development of muscular dystrophies [142]. Plectin 1b (P1b) was shown to be associated with mitochondria even after the fractionation of cells. Due to their localization [138,139], plectin isoforms 1b and 1d were proposed to be the most plausible candidates for the participation and control of desmin interconnections between IFs and OMM.

Plectin deficiency resulted in progressive degenerative changes in striated muscle, with a significant aggregation and partial loss of IFs, separation of the contractile system from sarcolemma, and alterations in the architecture of costameres. The decreased mitochondrial content was supported by lowered activity of mitochondrial matrix enzyme citrate synthase, frequently used as a mitochondrial marker [48]. Notably, mitochondria-rich oxidative muscles, like the heart and soleus, were most intensely influenced by plectin deficiency. Also, plectin 1b knockout models exhibited changes in mitochondrial shapes, without substantial alterations of mitochondrial motility [48]. In addition to muscles, in primary fibroblasts or myoblasts obtained from plectin-deficient mice, detachment of desmin IFs from Z-disks, costameres, mitochondria and nuclei, and formation of desmin aggregates were observed [140]. At the same time, the initial mitochondrial morphology can be partially restored by the overexpression of isoform-specific P1b in plectin-deficient muscles or in plectin-null cells. Furthermore, it has been demonstrated that some mutations of the plectin genes, or disturbance and pathological changes of the IFs, can be correlated with severe dysfunction of mitochondria [141,142,143]. Therefore, the effects observed earlier in desmin-deficient models (desmin-null cardiac and soleus muscles) can be explained by the specific participation of a mitochondrial isoform of the cytolinker protein plectin, connecting desmin to mitochondria. Depletion of plectin isoforms (P1b or P1d) significantly changed mitochondrial intracellular organization and respiratory function in conditional plectin knockout mice (MCK-Cre/cKO) [48]. Notably, morphological changes of mitochondria were associated with high levels of Mfn-2, a mitochondrial fusion protein. High resolution respirometry of permeabilized fibers from cardiac and skeletal muscles of conditional plectin knockout mice showed a reduction of maximal rates of respiration and significant decrease in the appKm for ADP, reflecting changed permeability of the OMM (the more open VDAC state) [48].

Taken together, these studies suggest the mitochondrial isoform of plectin 1b as a reliable candidate (similar to tubulin beta-II) for the regulation of mitochondrial respiratory function via its control of VDAC (Figure 5B).

## 8. Cytoskeletal-Mitochondria Interactions in Pathology

Injured mitochondria play important role in the mechanisms of a variety of pathological conditions, including muscular dystrophy, IRI and various cardiomyopathies. Functional disturbances of several cytoskeletal proteins, namely tubulin, plectin, desmin, and vimentin, can lead to various diseases, associated with the dysfunction of mitochondria and mitochondria-cytoskeleton connections/interactions [92].

Many mechanical interactions can be observed in the cell. Various mechanical forces are transmitted via the cytoskeleton to the different intracellular organelles like mitochondria, affecting their function and morphology. In cardiac cells, mitochondria may possess an evident mechano-sensitivity, serving as subcellular mechano-sensors and showing stretch-induced changes in their function (OXPHOS), Δψm, ROS and Ca^2+^ signaling. Such mechano-sensitivity of mitochondria may contribute to the mechanisms of several pathologies (heart failure, cardiomyopathies, arrhythmias, hypertension) including aging [144,145,146,147]. On the other hand, it has been shown that mitochondrial volume changes (e.g., swelling) may be mechanically transduced to the other cellular organelles like myofibrils and nuclei, altering their morphology and function [148].

The common and serious human genetic disease Duchenne muscular dystrophy (DMD) occurs due to the low expression of dystrophin [149]. Dystrophin is an important, tubulin binding, cytoskeletal protein, serving as cellular cytolinker and stabilizer of microtubules [150]. It is associated with glycoproteins of sarcolemma, connecting subsarcolemmal cytoskeleton with the extracellular matrix [151]. Dystrophin-deficient muscles demonstrated serious changes in mitochondrial physiology and function [152]. Mdx mice are frequently used as an animal model for DMD. It has been observed that, instead of well-organized microtubule lattice, DMD skeletal muscles showed largely disarranged microtubules. This was associated with the overexpression of the specific isoform of tubulin (beta 6 class V β-tubulin) pointing to the possible important role of this overexpression in DMD [153]. The study of mitochondrial bioenergetics demonstrated that, in permeabilized skeletal muscle fibers from mdx mice, the maximal rates of mitochondrial respiration were about twice lower than those of controls and similar changes were observed in skeletal muscle biopsies from DMD patients. It has been found that mitochondrial injuries were related to the damage to complex I of the respiratory chain, low creatine-stimulated respiration due to damage in MitCK, and therefore impairment of phosphocreatine shuttle, leading to a less efficient intracellular energy transfer. These findings show that the dysfunction of muscle mitochondria can be the beginning of cardiomyopathy observed in mdx mice [152]. Therefore, mitochondria can represent a target for the treatment of DMD-associated cardiomyopathies and the recovery of mitochondrial bioenergetics can be considered for DMD treatment in patients.

The cytoskeleton of cardiac cells represents a highly organized structure to transmit mechanical forces and maintain proper organization of cellular organelles. Significant concomitant changes in the cytoskeleton, mitochondria, and in their interrelations were observed in IRI. During ischemia, the disruption of the cytoskeleton and its components induces damage of the integrity of myocytes resulting in their destruction and loss [154,155,156,157]. Mitochondrial respiratory function and energy transfer via MitCK and PCr pathway also decreased in various cardiomyopathies, heart failure and after IRI [158,159,160,161]. Also, ischemia-associated cardiomyopathies may occur due to alterations in the expression and subcellular reorganization of various cytoskeletal proteins [162,163,164,165]. Notably, intracellular rearrangement and changes in the location of tubulin beta-II which, under normal conditions, is finely colocalized with mitochondria (see above), was observed after IRI in the Langendorff perfused rat heart model [155]. These changes were concomitant with the increased affinity for ADP in mitochondrial respiration and oxidative phosphorylation (decreased appKm for ADP) and accompanied by decreased functional coupling of these processes with MitCK [155]. The decrease of appKm for ADP in mitochondrial respiration was frequently found in cardiac IRI [166,167], demonstrating a reduction of micro-compartmentation effects and energy fluxes via coupled CK systems. Using this model, it was found that appKm for ADP in the control group and after IRI inversely correlated with left-ventricular end-diastolic pressure [155].

Tubulin beta-II can modulate mitochondrial permeability transition pore (PTP) opening during IRI. The PTP opening in a low conductance increases permeability to solutes ≤ 300 Da, mostly ions, and induces negligible matrix swelling which, in turn, stimulates ETC activity, OXPHOS, ATP production, fatty acid oxidation, and other metabolic processes [168,169]. However, a high-conductance PTP opening which occurs in response to oxidative stress such as cardiac IRI enhances unrestricted bi-directional movements of water and solutes ≤ 1500 Da across the IMM. As a result, excessive matrix swelling causes IMM depolarization, ATP depletion, and rupture of the OMM, leading to cell death (Reviewed in [170,171,172]). Several mitochondrial proteins such as cyclophilin D, VDAC, ANT, phosphate carrier, ETC complex I, and Bcl-2 proteins participate in the regulation of the PTP opening [173,174,175,176]. The interaction of tubulin beta-II with VDAC can change the PTP activity under physiological and pathological conditions. In favour of this, disrupting microtubule architecture in permeabilized muscle fibers that demonstrated direct interaction between α-tubulin and tubulin beta-II and VDAC2 decreased Ca^2+^ retention capacity due to increased PTP opening [177].

In the heart, Ca^2+^ is a main player in the control of excitation–contraction coupling, also playing an important role in mitochondrial bioenergetics and cytoskeleton functionality [178,179,180]. However, increased mitochondrial and cytosolic Ca^2+^ may lead to various pathologies and diseases, such as IRI, hypoxia-reoxygenation, arrhythmias, hypertension, heart failure, and metabolic syndrome, etc. [179,181,182,183,184]. This can be tightly associated with multiple Ca^2+^-modulated processes, such as: mitochondria/cell swelling, changes in the interactions between cytoskeleton, mitochondria, and SR, decline of ΔΨm, diminished mitochondrial respiratory capacity and consequent decreased cellular ATP content (energy stress), altered mitochondrial dynamics (fission-fusion balance, Drp1 signaling), increased ROS (oxidative stress), and induction of apoptosis [179,183,185]. It has been shown that a component of the cell-cell interactions (adhesion) machinery may be involved in the control of calcium cycling and homeostasis, and its deficiency may lead to the heart arrhythmia [182].

Interestingly, a heterogeneous value of appKm for ADP was found after relatively short periods of ischemia, revealing at least two populations of mitochondria with normal and low appKm for ADP [155]. Likewise, fluorescence confocal microscopy imaging revealed a heterogeneous response and damage of cardiac mitochondria in response to cold ischemia (organ cold storage, preservation) and reperfusion [186]. This can be due to the absence of electrical continuity of intermyofibrillar mitochondria that may prevent breakdown of the entire bioenergetics in the cell [187]. Furthermore, tubulin beta-II mitochondrial dislocation during IRI was comparable to that found in volume overload cardiac hypertrophy [188]. This dislocation might be due to protein degradation. A remodeling of the cytoskeleton (in particular the microtubular system) was discovered also in cardiac chronic hypertrophy [2,119], associated with increased beta-tubulin expression and tubulin C-terminus post-translational modifications [189,190,191,192], in heart failure and various cardiomyopathies [117,118,193].

## 9. Conclusions

Many structural and functional interactions were found to be involved in the integration of mitochondria with other cellular systems like the SR and cytoskeleton, connecting mitochondrial function, dynamics, and regulation with the entire cell physiology, in particular in the most energy consuming organ, the heart. Overall, existing studies provide strong evidence that cytoskeletal proteins such as tubulin beta-II and plectin 1b interact with mitochondria. The interaction regulates metabolic fluxes via the energy transferring supercomplex VDAC-MitCK-ANT which, in turn, coordinates mitochondrial respiration and OXPHOS and the entire cellular physiology. The detailed characterization of molecular mechanisms implicated in mitochondrial–cytoskeleton interactions under normal and pathological conditions can be helpful for the development of new therapeutic approaches.

It should be pointed out that many structural and functional aspects of mitochondria–cytoskeletal proteins interactions, as well as detailed molecular mechanisms of their formation, are not yet known, and the interactions of tubulin beta-II or plectin 1b with mitochondria (mitochondrial VDAC) have to be shown more directly, using the most modern methodologies, for example, by using: (i) imaging approaches with a higher levels of spatial and temporal resolutions, (ii) application of mitochondrial green fluorescent proteins (GFPs) specifically targeted to mitochondria, (iii) use of fluorescence resonance energy transfer (FRET) method to directly analyze possible protein-protein interactions and proximities, (iv) reconstruction or reconstitution experiments using tubulin β-II transfection and specific fragments of plectin, and (v) advancement of recombinant α- and β-tubulin isoforms with modifications of the C-terminal tail.

## Figures and Tables

**Figure 1 cells-09-00222-f001:**
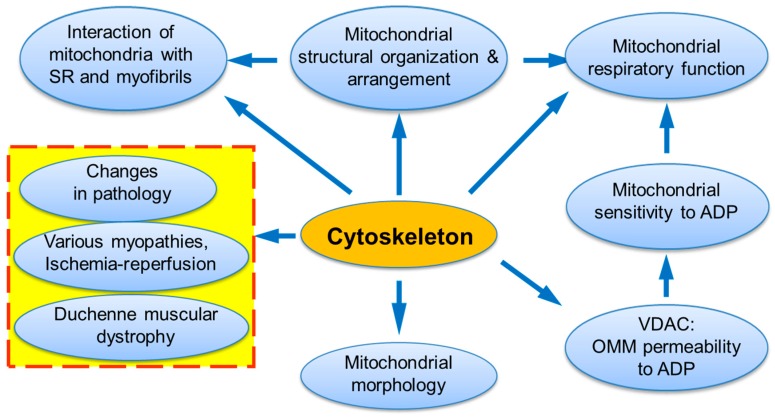
The central roles of cytoskeleton and its interactions in mitochondrial and entire cell physiology.

**Figure 2 cells-09-00222-f002:**
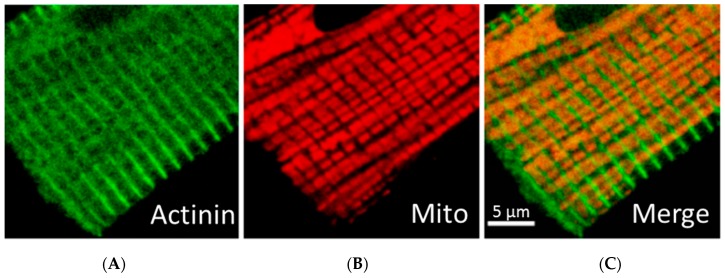
Simultaneous fluorescent and immunofluorescent confocal imaging analysis of mitochondria and sarcomeric Z-line (actinin) tubulin beta-II in rat cardiomyocyte. (**A**): Z-line (actinin); (**B**): Mitochondria; (**C**): Merge image. Scale bar, 5 µm.

**Figure 3 cells-09-00222-f003:**
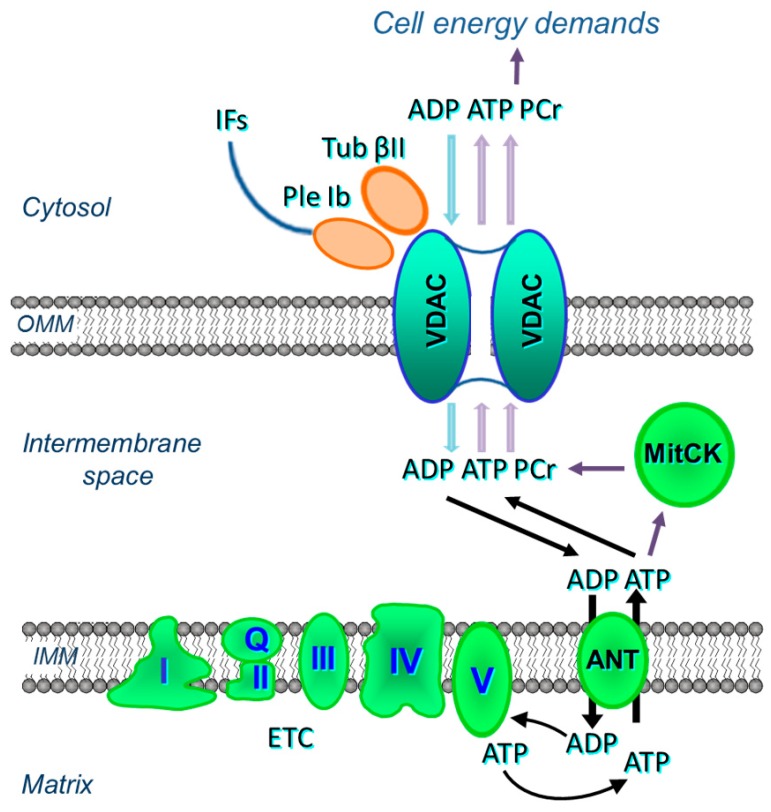
Possible interactions of VDAC of the OMM with tubulin beta-II (Tub βII), plectin 1b (Ple 1b), mitochondrial creatine kinase (MitCK) and ADP-ATP translocase (ANT) in cardiac cells.

**Figure 4 cells-09-00222-f004:**
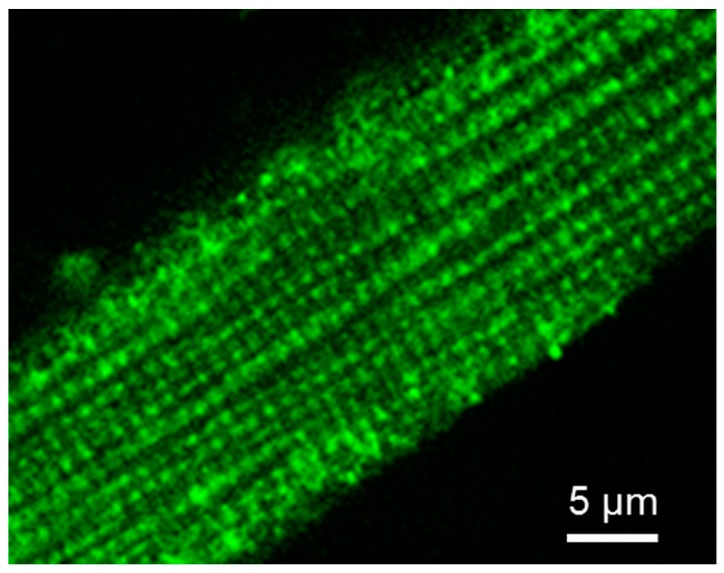
Cellular distribution of tubulin beta-II in adult rat cardiomyocyte. Tubulin beta-II was visualized by immunofluorescent confocal microscopy using specific antibodies. Scale bar 5 µm.

**Figure 5 cells-09-00222-f005:**
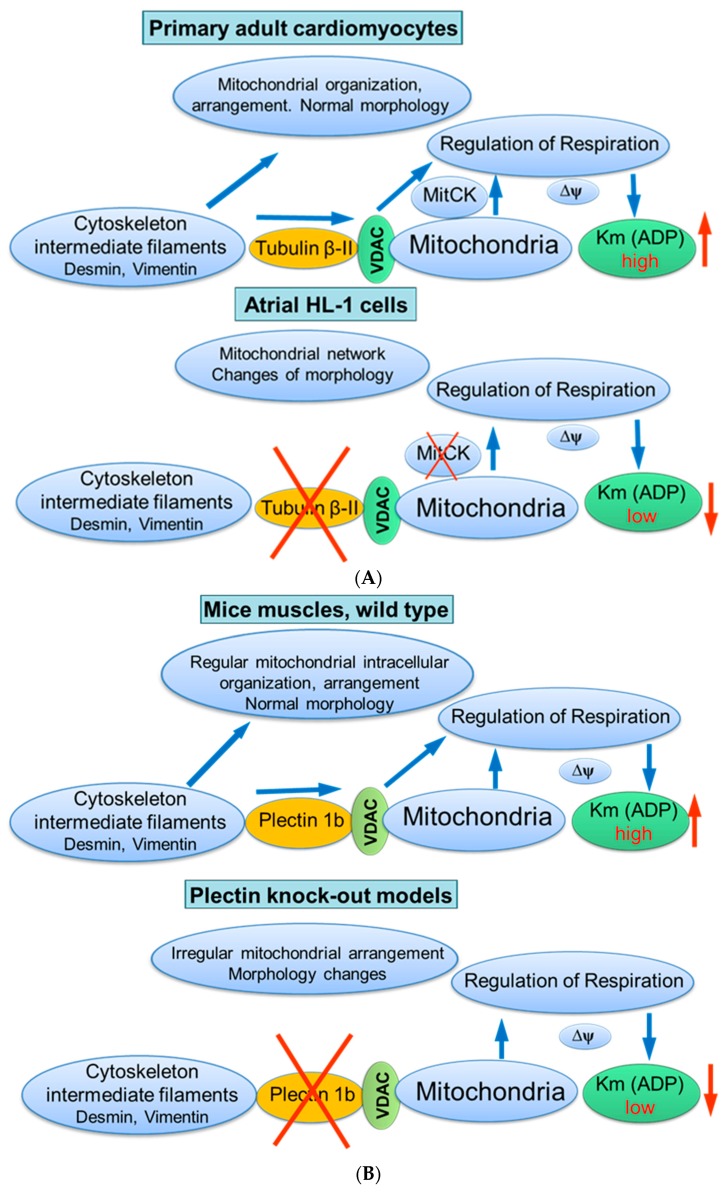
Possible role of cytoskeletal proteins tubulin beta-II (**A**) and plectin 1b (**B**) interactions with mitochondria. Scheme summarizing hypotheses regarding the control of mitochondrial respiratory function by tubulin beta-II and plectin 1b via their connections with VDAC. Km (ADP), apparent Michaelis constant for ADP; MitCK, mitochondrial creatine kinase; ΔΨm, the inner mitochondrial membrane potential.

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
