# Peer review of "Crosstalk between Mitochondria and Cytoskeleton in Cardiac Cells"

_cells, 2020, doi:10.3390/cells9010222_

Round 1
Reviewer 1 Report
In this review, Kuznetsov et al aim to summarize actual knowledge on the crosstalk between mitochondria and cytoskeleton in cardiac and muscle cells. The manuscript is potentially helpful. Additional figures and focus in the main text to augment key points would increase the value of this Review article. I do also recommend editing for proper English and typographical errors. With that, I recommend the following changes:
Title, “Intracellular crosstalk between mitochondria and cytoskeleton in cardiac and skeletal muscles” is for me confusing, as readers will expect a clear listing of what is know in cardiac versus other type of muscles (skeletal and/or smooth ones). However, no such differences are highlighted in the main text. Please give a real overview of the current knowledge of both mitochondria and cytoskeleton in the three main types of muscles, especially because those cells are completely different regarding formation and cytoskeleton organization (binucleated cells for cardiac cells, mononucleated cells for smooth muscle or multinucleated cells for skeletal cells), otherwise change the title and mention that you are just focusing on cardiac cells.
Cytoskeletons are usually classified in Actin, Microtubules and Intermediate filaments with their own set of associated proteins, and some of them binding several cytoskeletons. The Review will gain in clarity if we can easily access to the role of each cystoskeletons and their respective roles in mitochondria functionality, shaping, positioning.
Finally, it’s always a matter for me to identify some mechanisms in for example neurons cells or in yeast models and assume that it will follow the same mechanisms in muscle cells. I will appreciate to clearly mention when elusive where (models/cells) mechanisms has been find and to which extend we can apply it in muscle cells. For example, 158-159, in this part, the authors should mention that proteins cited are obtained in the yeast context! As far as I know, extrapolation in mammalian context is hazardous.
Ln147: Sarcomeric Z-lines are not T-tubules ! Intermyofibrillar mitochondria are normally organized at the M-band, exclude from the Z-line. Replace by “Accumulation of Mitochondria is observed at the vicinity of t-tubular network”.
Ln 511, Dystrophin is also a tubulin binding proteins (Prins 2009) and in DMD, up regulation of specific isoform is observed, pointing the role of other tubulin isoforms role in skeletal muscle (Randazzo et al 2018). Authors should comment.
Fig 4. Authors should show a co-staining of beta tub II and mitochondria
Fig5 &6: Authors should mix the two figures, as they are redundant.
Author Response
Reviewer 1
In this review, Kuznetsov et al aim to summarize actual knowledge on the crosstalk between mitochondria and cytoskeleton in cardiac and muscle cells. The manuscript is potentially helpful. Additional figures and focus in the main text to augment key points would increase the value of this Review article. I do also recommend editing for proper English and typographical errors. With that, I recommend the following changes:
Reviewer’s comment (RC): Title, “Intracellular crosstalk between mitochondria and cytoskeleton in cardiac and skeletal muscles” is for me confusing, as readers will expect a clear listing of what is known in cardiac versus other type of muscles (skeletal and/or smooth ones). However, no such differences are highlighted in the main text.
Authors’ Response (AR): Thank you for your comments and valuable suggestions that allowed us to improve the manuscript!
We agree with the Reviewer. Since we mostly focus on the adult cardiac cells, the title has been corrected to “Crosstalk between mitochondria and cytoskeleton in cardiac cells”. The differences between cardiac and other muscle types (in relation to the mitochondrial content, oxidative capacities, general energy metabolism, etc.) have been broadly studied previously. In particular, a drastic difference in mitochondrial bioenergetics has been shown between cardiac and smooth muscles (e.g. see Wiedemann et al. 2013).
RC: Please give a real overview of the current knowledge of both mitochondria and cytoskeleton in the three main types of muscles, especially because those cells are completely different regarding formation and cytoskeleton organization (binucleated cells for cardiac cells, mononucleated cells for smooth muscle or multinucleated cells for skeletal cells), otherwise change the title and mention that you are just focusing on cardiac cells.
AR: We agree with the Reviewer. Since we mostly focus on cardiac cells, the title has been corrected to - “Crosstalk between mitochondria and cytoskeleton in cardiac cells”.
RC: Cytoskeletons are usually classified in Actin, Microtubules and Intermediate filaments with their own set of associated proteins, and some of them binding several cytoskeletons. The Review will gain in clarity if we can easily access to the role of each cystoskeletons and their respective roles in mitochondria functionality, shaping, positioning.
AR: We mostly discuss the role of cytoskeleton in the regulation of mitochondrial function to follow certain requirements for review article including page limitation, and concentration on and discussion of certain topic in detail. We have already mentioned all these important points in the MS (see section 3).
RC: Finally, it’s always a matter for me to identify some mechanisms in for example neurons cells or in yeast models and assume that it will follow the same mechanisms in muscle cells. I will appreciate to clearly mention when elusive where (models/cells) mechanisms has been find and to which extend we can apply it in muscle cells. For example, 158-159, in this part, the authors should mention that proteins cited are obtained in the yeast context! As far as I know, extrapolation in mammalian context is hazardous.
AR: Thank you! This important point we have mentioned in the MS (Section 3; Ln: 190-192 “In contrast, in cardiac cells, mitochondria are strongly fixed between myofibrils, which is absolutely obligatory for the normal organ contractile function.” and 202-203 “Notably, mitochondrial morphology, intracellular arrangement, and specific proteins involved in the mitochondrial dynamics are extremely cell-tissue specific”). Information about yeast mitochondrial proteins has been removed from the MS.
RC: Ln147: Sarcomeric Z-lines are not T-tubules ! Intermyofibrillar mitochondria are normally organized at the M-band, exclude from the Z-line. Replace by “Accumulation of Mitochondria is observed at the vicinity of t-tubular network”.
AR: It has been corrected.
RC: Ln 511, Dystrophin is also a tubulin binding proteins (Prins 2009) and in DMD, up regulation of specific isoform is observed, pointing the role of other tubulin isoforms role in skeletal muscle (Randazzo et al 2018). Authors should comment.
AR: These studies are discussed and the references have been added.
RC: Fig 4. Authors should show a co-staining of beta tub II and mitochondria.
AR: A co-staining of beta tub II and mitochondria (imaging) has been published in our previous publication (“Antioxidants” 2019) and the article is cited as ref. 157.
RC: Fig 5 & 6: Authors should mix the two figures, as they are redundant.
AR: Thank you! The Figs. 5 and 6 have been combined as Fig. 5A,B.
Reviewer 2 Report
This is a very nice review made by Kuznetsov et al, focusing their attention on mitochondrial-cytoskeleton crosstalk in cardiac and skeletal muscle.
As I am not a expert for skeletal muscle I focussed my revision on the myocardial part.
Major: I think it is a well described review touching the majority of the aspect regarding the previous work and the work that need to be done in this field.
Recently sever authors proposed a novel role of mitochondria that, together with cytoskeleton act as mechanosensor, regulating the Calcium homeostasis in the cardiac tissue, via mechanoelectrical and mechanochemical feedback.
I think that a detailed paragraph may nicely complemented this review. In the light of this I would check the following articles:
Prosser BL, Ward CW, Lederer WJ. X-ROS signaling: rapid mechano-chemo transduction in heart. Science. 2011;333(6048):1440-5.
Rog-Zielinska EA, O'Toole ET, Hoenger A, Kohl P. Mitochondrial Deformation During the Cardiac Mechanical Cycle. Anat Rec (Hoboken). 2019;302(1):146-52.
Yaniv Y, Juhaszova M, Wang S, Fishbein KW, Zorov DB, Sollott SJ. Analysis of mitochondrial 3D-deformation in cardiomyocytes during active contraction reveals passive structural anisotropy of orthogonal short axes. PLoS One. 2011;6(7):e21985.
Miragoli M, Sanchez-Alonso JL, Bhargava A, Wright PT, Sikkel M, Schobesberger S, et al. Microtubule-Dependent Mitochondria Alignment Regulates Calcium Release in Response to Nanomechanical Stimulus in Heart Myocytes. Cell Rep. 2016;14(1):140-51.
Miragoli M, Cabassi A. Mitochondrial Mechanosensor Microdomains in Cardiovascular Disorders. Adv Exp Med Biol. 2017;982:247-64.
Iribe G, Kaihara K, Yamaguchi Y, Nakaya M, Inoue R, Naruse K. Mechano-sensitivity of mitochondrial function in mouse cardiac myocytes. Prog Biophys Mol Biol. 2017;130(Pt B):315-22.
Minors:
Figure 2: scale bar is not detailed.
Figure 4: scale bar is missing
Author Response
Reviewer 2
This is a very nice review made by Kuznetsov et al, focusing their attention on mitochondrial-cytoskeleton crosstalk in cardiac and skeletal muscle. As I am not a expert for skeletal muscle I focused my revision on the myocardial part.
Reviewer’s comment (RC): I think it is a well described review touching the majority of the aspect regarding the previous work and the work that need to be done in this field.
Recently sever authors proposed a novel role of mitochondria that, together with cytoskeleton act as mechanosensor, regulating the Calcium homeostasis in the cardiac tissue, via mechanoelectrical and mechanochemical feedback.
I think that a detailed paragraph may nicely complemented this review. In the light of this I would check the following articles:
Prosser BL, Ward CW, Lederer WJ. X-ROS signaling: rapid mechano-chemo transduction in heart. Science. 2011;333(6048):1440-5.
Rog-Zielinska EA, O'Toole ET, Hoenger A, Kohl P. Mitochondrial Deformation During the Cardiac Mechanical Cycle. Anat Rec (Hoboken). 2019;302(1):146-52.
Yaniv Y, Juhaszova M, Wang S, Fishbein KW, Zorov DB, Sollott SJ. Analysis of mitochondrial 3D-deformation in cardiomyocytes during active contraction reveals passive structural anisotropy of orthogonal short axes. PLoS One. 2011;6(7):e21985.
Miragoli M, Sanchez-Alonso JL, Bhargava A, Wright PT, Sikkel M, Schobesberger S, et al. Microtubule-Dependent Mitochondria Alignment Regulates Calcium Release in Response to Nanomechanical Stimulus in Heart Myocytes. Cell Rep. 2016;14(1):140-51.
Miragoli M, Cabassi A. Mitochondrial Mechanosensor Microdomains in Cardiovascular Disorders. Adv Exp Med Biol. 2017;982:247-64.
Iribe G, Kaihara K, Yamaguchi Y, Nakaya M, Inoue R, Naruse K. Mechano-sensitivity of mitochondrial function in mouse cardiac myocytes. Prog Biophys Mol Biol. 2017;130(Pt B):315-22.
Authors’ Response (AR): Thank you for your comments and valuable suggestions that allowed us to improve the manuscript! An additional paragraph and several new references including those mentioned by the reviewer have been added (Section 8, second paragraph).
RC: Figure 2: scale bar is not detailed.
AR: It has been corrected.
RC: Figure 4: scale bar is missing.
AR: Thank you! It has been corrected.
Reviewer 3 Report
The present review discusses the role played by different components of the cytoskeleton (microtubules, intermediate filaments, and microfilaments) in cardiac and skeletal muscle mitochondrial bioenergetics. The authors extensively describe and discussed the specific role of cytoskeletal proteins in the regulation of VDAC function/permeability. The text discusses the concept of energy channeling and more in detail how the supercomplex VDAC-MitCK-ADP-ATP can modulate mitochondrial bioenergetics. The proposed mechanism for the mitochondrial bioenergetics regulation is based on the reasonable hypothesis that the interaction of the cytoskeletal proteins tubulin beta II and plectin1b modulates VDAC permeability. This may play a critical role in mitochondrial physiology and pathology. This hypothesis is well supported by the previous work of the authors and other researchers on the field.
Minor comments:
Section 1, 2 and 3 are properly written following a nice flow. It starts with a clear introduction followed by a well-organized background (sections 1 and 2). Section 3 deepens on the role of the cytoskeleton in mitochondrial dynamics and its intracellular organization which is critical in highly organized cells as cardiomyocytes.
The cardiomyocytes host three different and well-differentiated mitochondrial populations. It would be of great interest to introduce the different cardiac mitochondrial populations in this particular topic. What is the role of the cytoskeleton in the distribution and dynamics of the subsarcolemmal, interfibrillar and perinuclear mitochondria in the cardiomyocytes? Is there any evidence of the cytoskeleton playing a critical role in bioenergetic peculiarities or the rearrangement of the different mitochondrial populations from the neonatal to adulthood stage?
Major concerns:
Sections 4 to 7:
These sections have a lack of flow. The sentences are extremely long and hard to read. The reader gets frequently lost due to the length of the sentences and lack of continuity in the proposed ideas.
Overuse of transition words, especially “Thus”. On many occasions the use of transitional words is unnecessary. The abuse of these grammatic element makes the text wordy. It is very difficult to get the ideas the authors want to express despite the expertise of the reviewer.
Please see below some examples and suggestions. A more detailed grammatical and structural correction is not viable due to the lack of time of the reviewer. Professional English proofreading is required for these sections.
Examples:
Section 4. SR/ER- mitochondrial interactions are introduced in line 211 but the concept of “mitochondria-associated membranes” does not appear in the text until the end of the section at line 231. Line 239: the subject of the verb is unclear. The absence of a clear subject often happens along these sections (4 to 7). Section 5 first paragraph. The concept of IFs properties, functions and its role in pathology are mixed and disorganized. The authors switch between IF and cytoskeleton without maintaining a clear order. Section 5. Second paragraph. After previously introducing IFs the authors start this paragraph talking about the cytoskeleton in general and its role in mitochondrial bioenergetics regulation. This first sentence seems out of place and should be introduced before. Desmin is introduced too late. The paragraph is extremely long. Concepts about function and role in pathology are mixed (jumpy). Please consider re-writing sentence starting at line 269 “For example, the absence of desmin in myocardial muscles…”. “Thus” is again overused, sometimes present twice in the same sentence. In general, the ideas seem jumpy and patchy. Difficult to find the flow. Line 298, spelling mistake “All”. Section 5 third paragraph. Line 290, the use of “suggested”, “potentially and “also” makes the sentence extremely wordy and complicated. It is difficult to get the idea of the authors and transmits uncertainty. The detailed explanation of the use of TMRE (line 295) is not necessary and looks out of context. Line 297. “Again” is not necessary for this sentence. Please consider that the sentence starting at line 297 and the following can be fused since they are redundant. Section 5 last paragraph. Please consider re-writing the sentence starting at line 305. Section 6. The new section starts abruptly. Consider restructuring this section by starting with the importance of VDAC permeability in the regulation of mitochondrial bioenergetics, suggesting the implication of tubulin in the permeability. This could followed by the description of tubulins and their role in the mitochondrial organization. Finally, the authors could link the previous ideas with a detailed discussion of the potential role of tubulins in VDAC permeability regulation. The sentence starting at line 311 is very long and must be cut. PLEASE, attention! The beginning of the sentences from line 316 and 321 are practically identical. Line 324: found also=> also found Line 326: “In the heart, tubulins form a network, which together with plectin and desmin, and with microfilament proteins (actin) creates precise structural organization of cardiac cells, obligatory for the organ contractile function, as well as for the regulation of required energy supply”. Very long sentence. Suggestion=> In the heart, tubulins form a network, which together with plectin and desmin, and microfilament proteins (actin) creates a precise structural organization of cardiac cells. This organization is essential to preserve the cardiac contractile function, as well as for the regulation of energy supply and demand. From line 326 to 339. Ideas are disconnected. The authors jump from tubulin polymerization states to the permeability of VDAC controlled by tubulin dimers. The concept of VDAC permeability controlled by tubulin will be described in detail two paragraphs later. Please pay attention to the structure and the flow. Line 340. The beginning of the sentence is very wordy and unnecessary. Line 346. It is striking to the reader that the authors wait until this line to point out that many metabolites (respiratory substrates, ADP, ATP and Pi) enter the mitochondria through VDAC. This sentence is out of place. Please, introduce it before. Line 374. This sentence starting the paragraph looks misplaced and not related to the second sentence. This arrangement cuts the flow and makes difficult to understand the idea that the authors want to communicate. Consider moving the sentence from line 374 to lines 382-387. From Line 374 to 440. The excessive use of transition words, “Again, additionally, In addition, notably, Importantly, Interestingly, Accordingly etc.” makes the text very wordy. These two paragraphs are easier to follow but they but seem disconnected from the previous paragraphs of the section. Please consider reorganizing the ideas of section 6 (previously suggested). Section 7. This section contains many concepts previously introduced. Please try to avoid repetitions. Try to be consistent with previous sections. Line 471. Mitochondrial respiration deficiency and defects in mitochondrial dynamics related to desmin deficiency have been introduced before. Line 476. “moreover” is not needed in the sentence. This sentence is too long and very difficult to read/understand. It must be rewritten.
No comments about section 8 and 9. These sections are nicely written and discussed. The pathological aspect of the cytoskeletal-mitochondrial interaction, especially its role in mPTP during IRI and its putative implication in cardiac remodeling, is well discussed and referenced.
Author Response
Reviewer 3
The present review discusses the role played by different components of the cytoskeleton (microtubules, intermediate filaments, and microfilaments) in cardiac and skeletal muscle mitochondrial bioenergetics. The authors extensively describe and discussed the specific role of cytoskeletal proteins in the regulation of VDAC function/permeability. The text discusses the concept of energy channeling and more in detail how the supercomplex VDAC-MitCK-ADP-ATP can modulate mitochondrial bioenergetics. The proposed mechanism for the mitochondrial bioenergetics regulation is based on the reasonable hypothesis that the interaction of the cytoskeletal proteins tubulin beta II and plectin1b modulates VDAC permeability. This may play a critical role in mitochondrial physiology and pathology. This hypothesis is well supported by the previous work of the authors and other researchers on the field.
Reviewer’s comment (RC): Section 1, 2 and 3 are properly written following a nice flow. It starts with a clear introduction followed by a well-organized background (sections 1 and 2). Section 3 deepens on the role of the cytoskeleton in mitochondrial dynamics and its intracellular organization which is critical in highly organized cells as cardiomyocytes.
The cardiomyocytes host three different and well-differentiated mitochondrial populations. It would be of great interest to introduce the different cardiac mitochondrial populations in this particular topic. What is the role of the cytoskeleton in the distribution and dynamics of the subsarcolemmal, interfibrillar and perinuclear mitochondria in the cardiomyocytes? Is there any evidence of the cytoskeleton playing a critical role in bioenergetic peculiarities or the rearrangement of the different mitochondrial populations from the neonatal to adulthood stage?
Authors’ Response (AR): Thank you for your comments and valuable suggestions that allowed us to improve the manuscript! This is very interesting point. Indeed, the different role of three subpopulations of mitochondria in healthy heart and cardiac diseases has been discussed broadly in previous studies (e.g. Kuznetsov et al. Antioxidants, 2019; Manneschi et al. J. Neurol. Sci. 1995; Lesnefsky et al. Am. J. Physiol 1997; Jimenez et al. Eur. J. Biochem. 2002; Dzeja et al. Proc. Natl. Acad. Sci. USA 2002 ; etc.). Unfortunately, there is practically no information about the role of cytoskeleton in the regulation of subsarcolemmal or perinuclear mitochondria that can be, at least partially, explained by the specific intracellular localization of these subpopulations. Here we were mostly focused on the intermyofibrillar mitochondria in adult cells.
RC: Sections 4 to 7: These sections have a lack of flow. The sentences are extremely long and hard to read. The reader gets frequently lost due to the length of the sentences and lack of continuity in the proposed ideas.
AR: In response to the reviewer’ comment, we have revised the text to improve its readiness and flow.
RC: Overuse of transition words, especially “Thus”.
AR: Thank you! We revised the MS to significantly reduce “Thus” in the text.
RC: On many occasions the use of transitional words is unnecessary. The abuse of these grammatical element makes the text wordy. It is very difficult to get the ideas the authors want to express despite the expertise of the reviewer.
Please see below some examples and suggestions. A more detailed grammatical and structural correction is not viable due to the lack of time of the reviewer. Professional English proofreading is required for these sections.
Examples: Section 4. SR/ER- mitochondrial interactions are introduced in line 211 but the concept of “mitochondria-associated membranes” does not appear in the text until the end of the section at line 231.
AR: It has been corrected.
RC: Line 239: the subject of the verb is unclear. The absence of a clear subject often happens along these sections (4 to 7).
AR: It has been corrected.
RC: Section 5, first paragraph. The concept of IFs properties, functions and its role in pathology are mixed and disorganized. The authors switch between IF and cytoskeleton without maintaining a clear order.
AR: It has been corrected.
RC: Section 5. Second paragraph. After previously introducing IFs the authors start this paragraph talking about the cytoskeleton in general and its role in mitochondrial bioenergetics regulation. This first sentence seems out of place and should be introduced before. Desmin is introduced too late. The paragraph is extremely long.
AR: It has been corrected.
RC: Concepts about function and role in pathology are mixed (jumpy).
AR: It has been corrected.
RC: Please consider re-writing sentence starting at line 269 “For example, the absence of desmin in myocardial muscles…”. “Thus” is again overused, sometimes present twice in the same sentence. In general, the ideas seem jumpy and patchy. Difficult to find the flow.
AR: It has been corrected.
RC: Line 298, spelling mistake “All”.
AR: It has been corrected.
RC: Section 5 third paragraph. Line 290, the use of “suggested”, “potentially and “also” makes the sentence extremely wordy and complicated. It is difficult to get the idea of the authors and transmits uncertainty.
AR: It has been corrected.
RC: The detailed explanation of the use of TMRE (line 295) is not necessary and looks out of context.
AR: It has been deleted.
RC: Line 297. “Again” is not necessary for this sentence. Please consider that the sentence starting at line 297 and the following can be fused since they are redundant.
AR: “Again” has been deleted.
RC: Section 5 last paragraph. Please consider re-writing the sentence starting at line 305.
AR: It has been deleted.
RC: Section 6. The new section starts abruptly. Consider restructuring this section by starting with the importance of VDAC permeability in the regulation of mitochondrial bioenergetics, suggesting the implication of tubulin in the permeability. This could followed by the description of tubulins and their role in the mitochondrial organization.
AR: It has been corrected.
RC: Finally, the authors could link the previous ideas with a detailed discussion of the potential role of tubulins in VDAC permeability regulation. The sentence starting at line 311 is very long and must be cut.
AR: It has been corrected.
RC: PLEASE, attention! The beginning of the sentences from line 316 and 321 are practically identical.
AR: It has been corrected.
RC: Line 324: found also=> also found
AR: It has been corrected.
RC: Line 326: “In the heart, tubulins form a network, which together with plectin and desmin, and with microfilament proteins (actin) creates precise structural organization of cardiac cells, obligatory for the organ contractile function, as well as for the regulation of required energy supply”. Very long sentence. Suggestion=> In the heart, tubulins form a network, which together with plectin and desmin, and microfilament proteins (actin) creates a precise structural organization of cardiac cells. This organization is essential to preserve the cardiac contractile function, as well as for the regulation of energy supply and demand.
AR: It has been corrected.
RC: From line 326 to 339. Ideas are disconnected. The authors jump from tubulin polymerization states to the permeability of VDAC controlled by tubulin dimers.
AR: It has been corrected.
RC: The concept of VDAC permeability controlled by tubulin will be described in detail two paragraphs later. Please pay attention to the structure and the flow. Line 340. The beginning of the sentence is very wordy and unnecessary.
AR: It has been corrected.
RC: Line 346. It is striking to the reader that the authors wait until this line to point out that many metabolites (respiratory substrates, ADP, ATP and Pi) enter the mitochondria through VDAC. This sentence is out of place. Please, introduce it before.
AR: We thought that this information might be directly connected to the role of tubulin in VDAC permeability and to the following Fig. 3.
RC: Line 374. This sentence starting the paragraph looks misplaced and not related to the second sentence. This arrangement cuts the flow and makes difficult to understand the idea that the authors want to communicate.
AR: It has been corrected.
RC: Consider moving the sentence from line 374 to lines 382-387. From Line 374 to 440. The excessive use of transition words, “Again, additionally, In addition, notably, Importantly, Interestingly, Accordingly etc.” makes the text very wordy.
AR: It has been corrected.
RC: These two paragraphs are easier to follow but they but seem disconnected from the previous paragraphs of the section. Please consider reorganizing the ideas of section 6 (previously suggested).
AR: It has been corrected.
RC: Section 7. This section contains many concepts previously introduced. Please try to avoid repetitions. Try to be consistent with previous sections.
AR: It has been corrected.
RC: Line 471. Mitochondrial respiration deficiency and defects in mitochondrial dynamics related to desmin deficiency have been introduced before.
AR: This sentence has been removed.
RC: Line 476. “moreover” is not needed in the sentence. This sentence is too long and very difficult to read/understand. It must be rewritten.
AR: It has been corrected.